# A Ridgeline-Based Terrain Co-registration for Satellite LiDAR Point Clouds in Rough Areas

**Ruqin Zhou** [1] and **Wanshou Jiang** [1,2,*]

1   State Key Laboratory of Information Engineering in Surveying, Mapping and Remote Sensing,
    Wuhan University, Wuhan 430079, China; zhouruqin@whu.edu.cn
2   Collaborative Innovation Center of Geospatial Technology, Wuhan University, Wuhan 430079, China
*   Correspondence: jws@whu.edu.cn; Tel.: +86-27-6877-8092 (ext. 8321)

**Abstract:** It is still a completely new and challenging task to register extensive, enormous and sparse satellite light detection and ranging (LiDAR) point clouds. Aimed at this problem, this study provides a ridgeline-based terrain co-registration method in preparation for satellite LiDAR point clouds in rough areas. This method has several merits: (1) only ridgelines are extracted as neighbor information for feature description and their intersections are extracted as keypoints, which can greatly reduce the number of points for subsequent processing, and extracted keypoints is of high repeatability and distinctiveness; (2) a new local-reference frame (LRF) construction method is designed by combining both three dimensional (3D) coordinate and normal vector covariance matrices, which effectively improves its direction consistency; (3) a minimum cost–maximum flow (MCMF) graph-matching strategy is adopted to maximize similarity sum in a sparse-matching graph. It can avoid the problem of "many-to-many" and "one to many" caused by traditional matching strategies; (4) a transformation matrix-based clustering is adopted with a least square (LS)-based registration, where mismatches are eliminated and correct pairs are fully participated in optimal parameters evaluation to improve its stability. Experiments on simulated satellite LiDAR point clouds show that this method can effectively remove mismatches and estimate optimal parameters with high accuracy, especially in rough areas.

**Keywords:** satellite LiDAR point clouds; local-reference frame; graph-matching; terrain registration

## 1. Introduction

Different from conventional two dimensional (2D) remote sensed imagery, light detection and ranging (LiDAR) can obtain dense point clouds with three-dimensional information of high accuracy. Over the past decades, with the hardware development and the progress of data processing algorithms, LiDAR has been carried on various platforms (airborne, unmanned aerial vehicles, vehicle, handheld) with wide applications, for example, electric power design [1], heritage modeling [2], water surveys [3], environment survey [4] and urban and land planning [5].

However, up to now, LiDAR has rarely been carried on satellites. There are two main obstacles: (1) the altitude of satellites is too high, and the distance between laser is certain, resulting in its low density; (2) the energy of laser emitter is limited, and a large percentage of energy is lost after long-distance transmission, resulting in its weak signal. In recent years, with the lightweight of laser sensors and the intensive density of point clouds, increasing attention has been poured into the satellite LiDAR data [6]. To improve the vertical accuracy of stereo satellite images without ground control, the ZY(Zi-Yuan)-302 satellite, launched on 30 May 2016, carried with it the first Chinese earth observation satellite laser altimeter, which can improve the vertical accuracy from 11 m to 1.9 m through active measurement [7]. ICESat (ice, cloud, and land elevation satellite)-2—launched on 15 September 2018, loaded a multibeam, non-scan, single-photon laser altimeter ATLAS (advanced topographic laser

altimeter system) for sea ice change detection, 3D surface information survey and global biomass estimation [4]. Global ecosystem dynamics investigation (GEDI), launched on 5 December 2018 with a full-waveform LiDAR, was designed to provide the first global, high-resolution observations of forest vertical structure [8]. With the successive launch of satellites equipped with lasers, the processing of satellite LiDAR data exerts a growing interest among researchers [9–11]. Registration of satellite LiDAR point clouds with sparse density and large footprint is a big challenge, and to date, there is no relevant research. With the registration algorithms of satellite images are increasingly mature in multi-spectral [12], multi-temporal [13] and multisensor [14], it's provide a good example for satellite LiDAR point cloud registration. Thus, to address its challenge, this study proposes a ridgeline-based terrain co-registration of satellite LiDAR point clouds.

### 1.1. Related Work

As one of the crucial steps in 3D point cloud processing, registration has two main phases: the coarse alignment and the fine alignment [15]. For decades, except ICP (iterative closest point)-based fine registration, a multitude of coarse registration algorithms have emerged to transform the 'source' point clouds to the 'target' point clouds. These methods can be categorized into three types: RANSAC (random sample consensus)-based methods, 4PCS (4 points congruent sets)-based methods and LS (least square)-based methods.

#### 1.1.1. RANSAC and Its Variations

RANSAC [16], as one of widely used strategies to cope with noise in point cloud registration [17], is a random, data-driven process with two iterative steps: generating a hypothesis by random samples, and verifying hypotheses by the remaining data [18]. However, it is of low efficiency and accuracy when there is a host of outliers.

To address these challenges, many variations have been developed. Considering that the distance between inliers is closer than that between outliers, adjacent points sample consensus (NAPSAC) [19] algorithm was proposed to handle datasets with high dimension and low proportion of inliers. To improve the registration efficiency, the progressive sample consensus (PROSAC) [20] took initial matching results as a basis of sorting, and sampled the results from high to low. In ARRSAC (adaptive real-time random sample consensus) [21], more than one model was generated, and the best model was selected by iteratively sorting and choosing according to scores of an objective function. GroupSAC [22] combined both NAPSAC and PROSAC, where points were grouped according to similarities and then sampled from the largest clustering according to the PROSAC. However, a prior knowledge was needed for classification. Persad et al. [15] proposed a threshold-free modified-RANSAC to eliminate false keypoint matches, where a spatial nearest neighbor consistency check was applied to ensure that true correspondences were stored as inliers. Quan et al. [23] introduced an efficient and accurate GC1SAC (global-constrained one-point-based sample consensus) algorithm for robust transformation estimation, where only one correspondence and its LRF were utilized for iterative computation.

#### 1.1.2. PCS and Its Variations

The 4PCS [24] algorithm was developed under the RANSAC framework. Through constructing and finding similar 4-point congruent sets in the target and source point clouds, it could achieve satisfactory registration results even with small overlap and noise. The affine invariance constraint and the LCP (large common point set) strategy were used to find 4-point congruent sets with maximum overlap. However, it was of huge search space, high complexity and low efficiency.

To speed up, a series of improvements based on 4PCS came into being. In K4PCS (keypoint 4PCS, [25]) algorithm, a keypoint detection was added instead of random sampling to achieve high registration accuracy. Super 4PCS [26] algorithm adopted an intelligent index strategy to reduce the computational complexity from $O(n2)$ to $O(n)$. The algorithm could be used for point cloud registration of large areas, but it had a bottleneck while searching coplanar points in all available source points.

The generalized 4PCS [27] algorithm improved the construction process by generalizing the 4-point base, which was no longer strictly limited in a plane. It greatly improved the matching efficiency. Super generalized 4PCS [28] algorithm was a combination of super 4PCS algorithm and generalized 4PCS algorithm, using an intelligent index strategy and non-coplanar optimization. Semantic keypoint 4PCS [29] was proposed aiming at urban area registration, where semantic keypoints of building objects were extracted layer by layer, instead of original random sampling points for point cloud registration. The 2PNS (2 point + normal sets) [30] algorithm used normal vectors between two points to build matching pairs. It greatly speeded up and can cope with the scene registration with a small overlap. To reduce the number of 4-point sets, Xu et al. [31] proposed a multiscale sparse features 4PCS (MSSF4PCS) algorithm, adding a normal vector constraint for point-matching. At the same time, it optimized matching results and improves registration accuracy by calculating point characteristics in the neighborhood of 4-point pairs. In V4PCS (volumetric 4PCS) [32,33] algorithm, volume consistency was added to construct the coplanar four-point base, extending coplanar four-point-matching to non-coplanar four-point-matching. It could reduce computational complexity and improves operational efficiency.

### 1.1.3. LS and Its Variations

Least square (LS) is a mathematical optimization strategy, finding the best function of data by minimizing the square sum of errors. However, this method is sensitive to noise and when the fitting function is nonlinear, iteration is required.

Originally, the LS algorithm was widely used for 2D image alignment. Due to its high flexibility and powerful mathematical model [34], a large amount of LS-based methods was put forward to enhance its robustness. Gruen and Akca [34] used the LS-based method for surface-matching based on a generalized Gauss–Markov (GM) model. However, only random errors of target point clouds were modeled in its observation equations. A constrained total least squares (CTLS) was introduced by Chen et al. [35] to solve large rotation registration with an errors-in-variables (EIV) model. Compared with traditional constrained least squares (CLS) methods, it could well solve the impact of random errors. Aydar et al. [36] considered stochastic properties of both observations and parameters and proposed a total least square (TLS) registration of 3D surfaces, where an EIV model was utilized and its parameters were estimated by a TLS method. Ge and Wunderlich [37] proposed a surface-based-matching of 3D point clouds with a nonlinear Gauss–Helmert model to solve the weighted total least-squares problem. The Gauss–Helmert least-squares 3D (GH-LS3D) method considered random errors of both target and source. Ahmed et al. [38] proposed a least-squares registration of point sets over SE(d) (subgroup of Euclidean group E(d)) using closed-form projections, where the original least-squares problem was derived from the SDP (semidefinite program) solution via a linear map, solved by a variable splitting and ADMM (alternating directions method of multipliers). To avoid the influence of gross errors and extend its application, Yu et al. [39] proposed an advanced outlier detected total least-squares (ODTLS) method. However, the ODTLS was a dual iterative solution, induced to huge computation.

### 1.2. Contribution

As one of the important topographic feature lines, ridgelines play a vital role in terrain skeleton structure representation, forming topography boundaries together with valley lines [40]. They have two special advantages: stability, and distinctive when point clouds are sparse. Thus, this paper proposes a ridgeline-based terrain co-registration in preparation for satellite LiDAR point clouds in rough areas. The main contribution of the paper manifests in four aspects:

(1) for extensive and enormous raw data, only point clouds of ridgelines are extracted as neighbor information for feature description, and then their intersections are regarded as keypoints. This step can efficiently reduce the number of points for subsequent processing and extracted keypoints is of high repeatability and distinctiveness.

(2) for the direction instability of the LRF's X and Y axes decomposed by traditional coordinate covariance, this study designs a new LRF construction method by combining both 3D coordinate covariance and normal vector covariance, which can effectively improve its direction consistency.

(3) the descriptor-matching is converted into a graph-matching, where a minimum cost–maximum flow method is adopted. It can efficiently achieve a global optimum in a sparse-matching graph and avoid the problem of "many-to-many" and "one to many" caused by traditional nearest neighbor-based or ratio-based matching strategies.

(4) different from traditional RANSAC or 4PCS-based methods, the registration makes full use of correct correspondences, where a transformation matrix-based clustering method is applied to reject gross error and select the maximum similar sample set and then optimal parameters are solved by the Rodriguez-LS-based algorithm. This step can effectively eliminate mismatches and obtain optimal parameters with high accuracy.

### 1.3. Overview

The paper is organized as follows. Section 2 introduces three kinds of DEM to analog satellite LiDAR point clouds in experiments. Section 3 presents the details of the ridgeline-based terrain co-registration method for satellite LiDAR point clouds. In Section 4, criteria for accuracy analysis are introduced. Experimental results and evaluations of the proposed method are shown in Section 5. Section 6 discusses the effect of different overlap, topography and noise on the proposed co-registration algorithm. Finally, a conclusion and some future researches related to this study are presented in Section 7.

## 2. Datasets

It is noteworthy that since there is no satellite LiDAR point cloud up to now, we use existing worldwide DEM (digital elevation model) data and airborne LiDAR point clouds to simulate satellite LiDAR point clouds. In the following experiments, three kinds of worldwide DEM data are utilized: ALOS DEM, ASTER GDEM and SRTM DEM, where the point distance of generated point clouds is equal to the resolution the original DEM.

**ALOS DEM:** it is an acquisition of the advanced land observing satellite (ALOS), launched in 2006, with a phased array L-band synthetic aperture radar (PALSAR). It has three observation modes: high resolution, scanning synthetic aperture radar and polarization. The horizontal and vertical accuracy of the DEM can reach 12 meters. The data were downloaded from the website: http://www.eorc.jaxa.jp/ALOS/en/aw3d30/data/index.htm.

**ASTER GDEM:** it is a global digital elevation data jointly released by the American National Aeronautics and Space Administration (NASA) and the Japanese Ministry of Economy, Trade and Industry (METI). The DEM is based on the observation results of Terra, a new generation of earth observation satellite of NASA. It is made of 1.3 million stereo image pairs collected by ASTER (advanced spaceborne thermal emission and reflection radiometer) sensors, covering all land areas between 83°N and 83°S. Its horizontal accuracy is 30 m with 95% confidence, and the elevation accuracy is 20 m with 95% confidence. The data were downloaded from the website: http://reverb.echo.nasa.gov/reverb/.

**SRTM DEM:** it is provided by NASA's shuttle radar topographic mission (SRTM), covering areas between 60°N and 60°S. According to its accuracy, SRTM DEM can be divided into SRTM1 and SRTM3, with the corresponding resolution of 30 m and 90 m, respectively. The SRTM3 DEM is available for over 80% of the globe. SRTM1 DEM was also produced but is not available for all countries. The resolution of the DEM data used in the following experiments is 90 m. The vertical accuracy of the DEM is reported to be less than 16 m. The data were downloaded from the website: http://srtm.csi.cgiar.org/download.

**IN 2011–2013:** 2011–2013 Indiana statewide LiDAR, provided by the OpenTopography facility with support from the National Science Foundation under NSF Award Numbers 1833703, 1833643 and 1833632. Ownership of the data products resides with the State of Indiana. All Lidar, DEM

and ancillary data products produced through this project are public domain data. The data were downloaded from the website: https://portal.opentopography.org.

As shown in Figure 1, in the experiments, we mainly used the following six datasets, where Figure 1a–c is with different resolutions in the same region and Figure 1d–f are of small overlap with the same resolution. For the first three data (Figure 1a–c)—although their resolutions and accuracy are different—trends of their topography are basically coincident from a macro perspective. From the microscopic view (Figure 1f,g), with the decreasing resolution, their details become poor and blurred, which only reflects the overall distribution and many small cliffs were ignored. Figure 1d,e is simulated LiDAR point clouds in flat and rough terrain, where Line 1 and Line 2 (Figure 1d), Line 3 and Line 4 (Figure 1e) are only partial overlapped. Their detailed information are reported in Table 1.

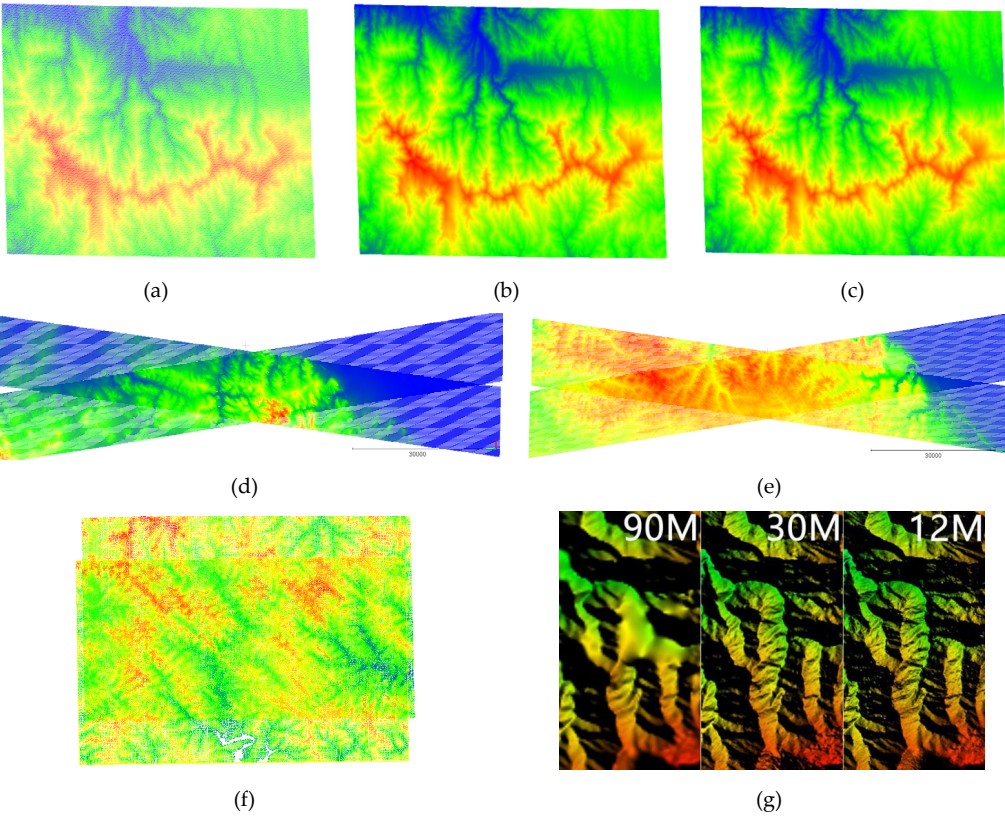

(a)  (b)  (c)

(d)  (e)

(f)  (g)

**Figure 1.** Datasets used in experiments. (**a**–**c**) are the simulated point clouds of SRTM 90, ASTER 30 and ALOS 12, respectively, which have different resolutions in the same region; (**d**,**e**) are simulated LiDAR point clouds in flat and rough terrain, where Line 1 and Line 2 (**d**), Line 3 and Line 4 (**e**) are only partial overlapped; (**f**) is the simulated light detection and ranging (LiDAR) point clouds from airborne LiDAR point clouds and (**g**) are detail comparison of three datasets in DEM and contours, respectively.

**Table 1.** Information on simulated LiDAR point clouds.

| Figure 1 | Dataset | Area | Vertical Resolution | Parallel Resolution | Points |
|---|---|---|---|---|---|
| (a) | SRTM 90 | 58,281 m * 57,505 m | 90 m | 90 m | 226,426 |
| (b) | ASTER 30 | 58,281 m * 57,505 m | 30 m | 30 m | 2,037,003 |
| (c) | ALOS 12 | 58,281 m * 57,505 m | 12 m | 12 m | 12,388,722 |
| (d) | Line 1 | 17,000 m * 60,000 m | 2 m | 120 m | 7,418,831 |
| | Line 2 | 17,000 m * 60,000 m | 2 m | 120 m | 7,418,831 |
| (e) | Line 3 | 17,000 m * 60,000 m | 2 m | 120 m | 7,445,621 |
| | Line 4 | 17,000 m * 60,000 m | 2 m | 120 m | 7,445,621 |
| (f) | IN-1 | 15,483 m * 24,341 m | 1 m | 1 m | 150,274,036 |
| | IN-2 | 15,638 m * 25,353 m | 1 m | 1 m | 158,146,927 |

* It is worth noting that although the resolutions of the above data are different, they were first downsampled to 120 m in the experiment to make them sparser.

## 3. Methodology

As shown in Figure 2, similar to a majority of descriptor-based registration methods, the proposed ridgeline-based terrain co-registration for satellite LiDAR point clouds is composed of several steps: ridgeline-based keypoint extraction, stable LRF construction, voxel descriptor generation, MCMF-based feature-matching, transformation matrix-based clustering and Rodriguez-LS-based registration, which are respectively introduced below.

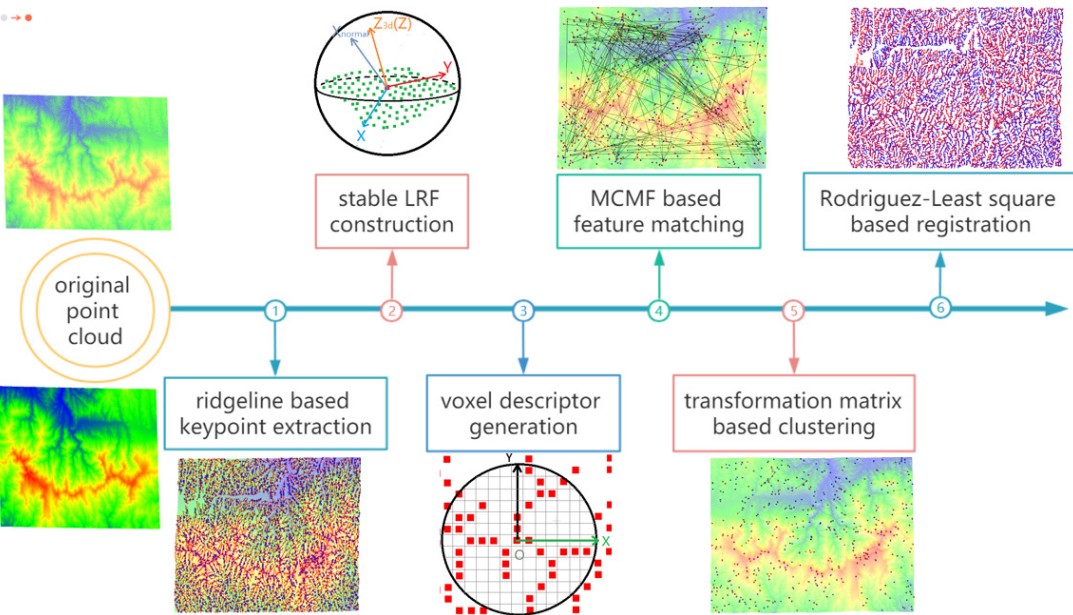

**Figure 2.** Workflow of the proposed method.

### 3.1. Ridgeline-Based Keypoint Extraction

Satellite LiDAR point clouds are extensive, enormous and sparse and its feature varies slightly on a local surface. Ridgelines, as one of the important macroscopic topographic feature lines [41], are stable and distinctive when point clouds are sparse. Thus, this study designs a ridgeline-based keypoint extraction method.

As shown in Figure 3, to reduce the number of points for subsequent processing, the original point clouds are downsampled and divided into grids first (the sampling distance and grid size are both set to 120 m in this study). Second, a mean filter with a window (size is set to $5*5$) is conducted on the sampled data. By subtracting the averaged data from the sampled data, the outlined data are obtained. Then, to get ridgelines, outline point clouds are refined by an improved thinning method. By summarizing previous classical thinning algorithm [42], thinning rules for each iteration are defined as follows: (1) the deleted point must be the foreground point; (2) the deleted point must be the edge point; (3) the endpoint cannot be deleted; (4) the isolated point cannot be deleted; (5) the connectivity should be unchanged; (6) the edge should be deleted first. After refinement, a spatial clustering [43] is utilized to remove trivial points. Finally, intersections with ridgelines in at least three directions are extracted as keypoints.

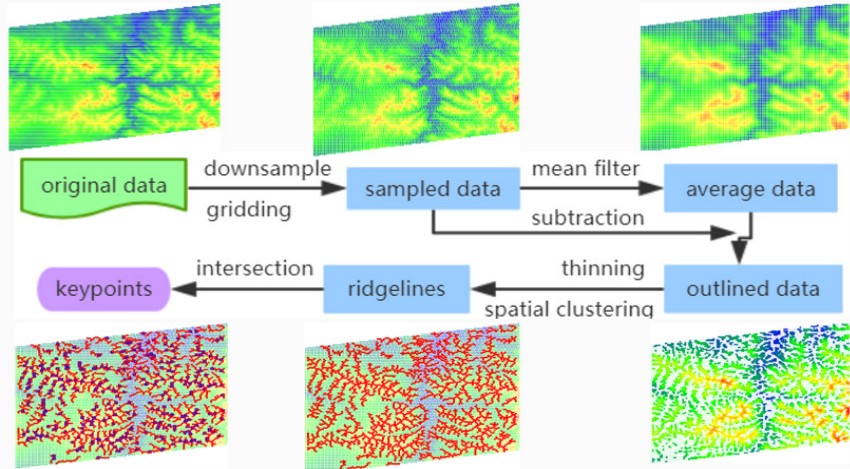

**Figure 3.** Ridgeline-based keypoint extraction.

## 3.2. Stable LRF Construction

A unique and stable LRF plays a significant role in both a descriptor's robustness and descriptiveness [44]. A descriptor's invariance to rigid transformation can be achieved by its LRF. The core of LRF construction is to determine two stable axes, which is traditionally determined by a coordinate or normal vector covariance matrix. Usually, the maximum eigenvector is chosen as the X-axis, while the minimum is the Z-axis [23,44,45]. However, the direction of these LRF is ambiguous and through experiments (presented in Section 5.2), it is found that the minimum eigenvector decomposed by 3D coordinate covariance matrix is relatively more stable and unique than other two eigenvectors (Figure 4a), while the maximum eigenvector decomposed by normal vector covariance matrix is the same (Figure 4b). Therefore, inspired by [46], the 3D coordinate covariance and normal vector covariance are combined to construct a stable LRF.

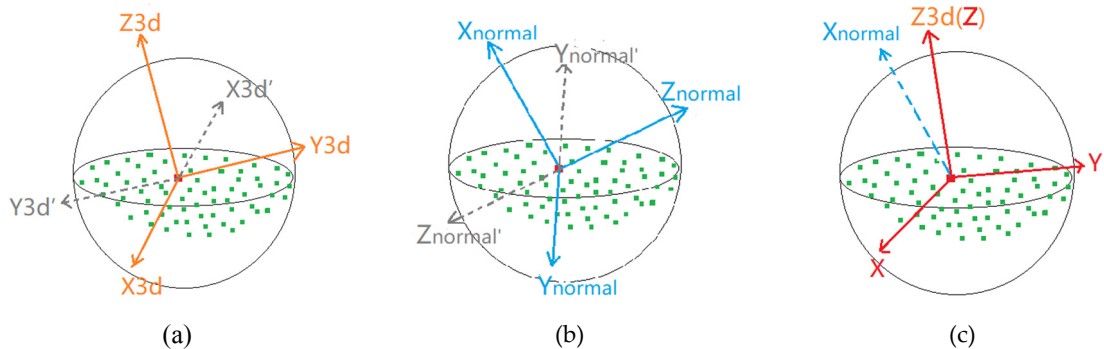

|  (a) |  (b) |  (c) |

**Figure 4.** Local-reference frame (LRF) construction. (**a**,**b**) are, respectively the LRF constructed by coordinate covariance matrix and normal vector covariance matrix, (**c**) is the proposed LRF combining coordinate and normal vector covariance.

Once per keypoint $p_i$ is detected, neighbor points $q_j$ in the local surface $S_i = \{q_j, \| q_j - p_i \| < R\}$ are obtained by a radius $R$, where $\|q_j - p_i\|$ is the distance between keypoint $p_i$ and neighbor points $q_j$. Moreover, their 3D coordinate $c_j$ $(x, y, z)$ and normal vector $n_j$ $(a, b, c)$ are, respectively used to construct the 3D coordinate covariance (Equation (1)) and the normal vector covariance (Equation (2)). Then, a PCA (Principal Component Analysis) decomposition is, respectively used for two covariances to acquire their eigenvectors. Let the minimum eigenvector of the 3D coordinate covariance be $Z_{3d}$ and the maximum eigenvector of the normal vector covariance be $X_{normal}$. The LRF is defined as

Equations (3)–(5), where $c_i$, $n_i$ are the coordinate and normal vector of keypoint $p_i$. To remove the ambiguity of LRF, the $X_{normal}$ and $Z_{3d}$ axes are oriented towards the majority direction of vectors [47].

$$C = \frac{1}{\sum\limits_{q_j \in S_i} (R - \|q_j - p_i\|)} \sum_{q_j \in S_i} (R - \|q_j - p_i\|)(c_j - c_i)(c_j - c_i)^T \tag{1}$$

$$C = \frac{1}{\sum\limits_{q_j \in S_i} (R - \|q_j - p_i\|)} \sum_{q_j \in S_i} (R - \|q_j - p_i\|)(n_j - n_i)(n_j - n_i)^T \tag{2}$$

$$Z = Z_{3d} \tag{3}$$

$$X = Z \times X_{normal} \tag{4}$$

$$Y = Z \times X \tag{5}$$

### 3.3. Descriptor Generation

Voxelization is an outstanding surface representation strategy with high efficiency in both computation and storage [48]. It has been extensively utilized in robotics and computer vision field [23]. Considering point clouds of ridgelines with wide footprints and sparse density, 3D voxelization is of high redundancy. Thus, a 2D rasterization is applied for feature description.

Once the LRF of each keypoint $p_i$ is established, the local surface $S_i$ is first transformed into a new coordinate system according to its LRF to keep rotation and translation invariance. Considering that the Z-axis of LRFs is more stable compared with its X and Y axes, the transformed local surface $S_i'$ is projected onto the XY plane of the LRF (Figure 5a). Then, the projected plane $S_p$ is divided into 2D rasters with the number of $g * g$ (Figure 5b). Further, a raster is coded according to whether it contains points or not. As shown in Figure 5c, if the raster contains points, it is coded as '**1**', otherwise, it is '**0**'. Thus, far, after all rasters in the neighborhood are coded, the final descriptor of the keypoint $p_i$ is generated by integrating all labels into a bit string with a length of $g * g$ (for example, {000000010000......0000000100000}).

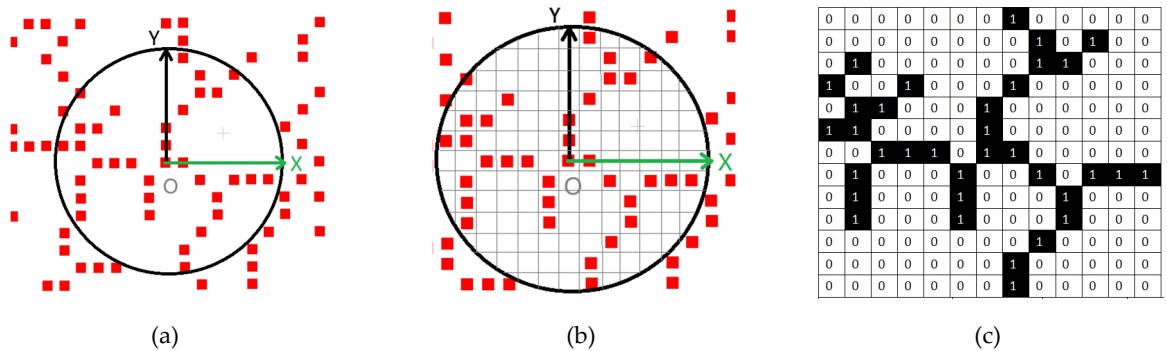

|     (a)     |     (b)     |     (c)     |

**Figure 5.** Descriptor generation. (**a**), (**b**) and (**c**) are sketch map of transformed local surface, rasterization and coding.

### 3.4. Max Flow Min Cost-Based Graph-Matching

Similar to our previous work [45], after the descriptor generation, we define the length of the descriptor minus Hamming distance as the similarity score, where the Hamming distance is the number of '1' in x XOR y (x and y are binary descriptors). Following, a feature-matching strategy is required. Traditional matching strategies, such as nearest neighbor (NN) or nearest neighbor distance ratio (NNDR)-based strategies, tend to fall into a local optimum [49]. The global strategy, KM (Kuhn_Munkres) algorithm adopted in our previous work [45], has a shortcoming that its matching

matrix must be square. However, factually, the amount of keypoints in the target and source is not equal, and matches with small similarities usually do not need to be considered. Thus, an efficient matching strategy called the minimum cost–maximum flow (MCMF) [50] is introduced for the non-square sparse matrix.

**Graph construction:** as shown in Figure 6, the capacity cost graph $G = (V, E, c, w)$ in MCMF algorithm is constructed as follows: the graph $G$ has $N$ nodes which include one virtual source node **vs**, one virtual sink node $v_t$, one layer $v_i$ ($1 \leq i < m$) corresponding to keypoints in the source and one layer $v_j$ ($m \leq j < N$) corresponding to keypoints in the target. For arcs $e = (vs, v_i)$ or $e = (v_j, v_t)$ connected to the source node **vs** or sink node $v_t$, their capacity $c_{si}$ or $c_{jt}$ are set as 1 and their cost $w_{si}$ and $w_{jt}$ of unit flow is 0. For other arcs $e = (v_i, v_j)$, their capacity $c_{ij}$ is set as 1, while their cost of unit flow $w_{ij}$ is set as 1-$s_{ij}$ ($s_{ij}$ is the similarity between the keypoint $v_i$ and keypoint $v_j$). Let $f$ be a feasible flow on capacity cost network $G$, the cost of the flow $f$ is defined as $w(f) = \sum_{(v_i, v_j) \in E} w_{ij} f_{ij}$, where $f_{ij}$ is the actual flow on each arc $e = (v_i, v_j)$.

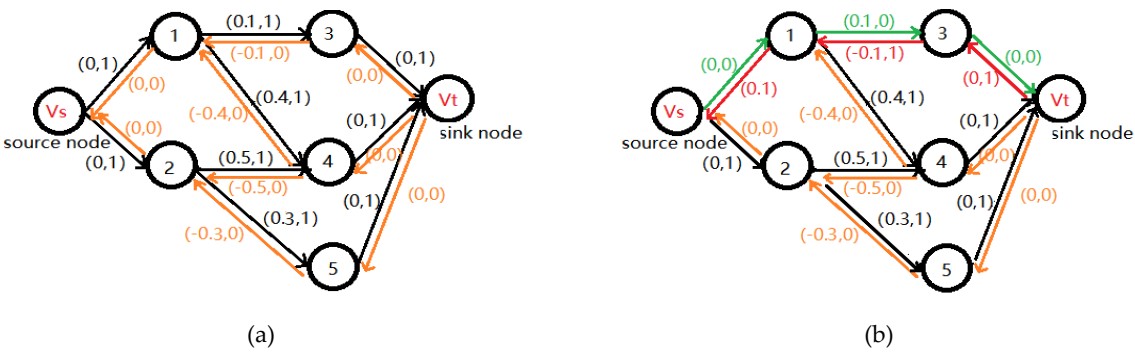

(a)                                        (b)

**Figure 6.** Minimum cost–maximum flow (MCMF)-based graph-matching. (**a**,**b**) are, respectively the residual digraph construction and update.

**Graph-matching:** after the capacity cost graph $G$ is constructed, a minimum cost–maximum flow (MCMF) strategy is adopted to find a feasible maximum flow $f = \{ f_1, f_2, f_3, \dots , f_k \}$ with $k$ matching pairs ($k$ is a given non-negative value) and total minimum cost $w(f)$. In each iteration of the augmented path searching, a weighted residual digraph (Figure 6a) is constructed according to the current cost flow graph $G$. Each arc $e = (v_i, v_j)$ in the capacity cost graph is decomposed into two mutually inverse arcs $v_{ij}$ (colored in black in Figure 6a) and $v_{ji}$ (colored in orange in Figure 6a). The capacity and cost of inverse arcs are defined as follows, where its capacity $c_{ji} = 1 - c_{ij}$ and its cost $w_{ji} = -w_{ij}$. Through each flow $f_{ij}$, the correspondence $(v_i, v_j)$ can be obtained. The detailed steps are shown in the Appendix A, Table A1.

### 3.5. Transformation Matrices-Based Clustering

After graph-matching, there are still a large number of mismatches (shown in Figure 7a). However, in previous literature [15,23,24,26], RANSAC and 4PCS-based algorithms were frequently used to deal with noise, which do not make full use of all correct matching pairs. Thus, considering that the correct matches' transformation matrix is of high consistency, while the wrong ones are diverse, a transformation matrix-based clustering is applied to eliminate mismatches.

#### 3.5.1. Transformation Matrix Computation

Theoretically, if only 3D coordinates of correspondences are used, at least three correspondences are required to solve a transformation matrix between two 3D models. Raposo and Barreto [30] calculated the transformation matrix by two correspondences and their normal vectors. It is noteworthy that, if the LRF of keypoints has been established, only one correspondence is adequate [23]. Therefore,

we use one correspondence and its LRF to calculate its transformation matrix. The equation is shown as follows:

$$R_i = L_i^T (L_i^S)^{-1} \tag{6}$$

$$T_i = q_i^T - R_i p_i^S \tag{7}$$

where $R_i$ is the rotation matrix, $T_i$ is the translation matrix, $p_i^S$ and $q_i^T$ is a correspondence in the model and scene. $L_i^S$ and $L_i^T$ are LRFs, respectively located at keypoints $p_i^S$ and $q_i^T$.

### 3.5.2. Maximum Sample Set Chosen by Clustering

Traditional RANSAC-based methods are commonly applied to remove mismatches, which are time-consuming and sensitive to high outlier ratios. Considering that correct matches are of high consistency, while wrong ones are diverse, Yang et al. [51] proposed a consistency voting (CV) method, however, an initial voting set should be predefined. Inspired by that, a transformation matrix-based clustering is adopted to eliminate mismatches without a predefined initial voting set.

It is clearly shown in Figure 7a that transformation matrices of correct correspondences are always consistent with each other (especially translation matrix), whereas incorrect correspondences are rarely compatible with either correct or incorrect ones. Thus, two consistent requirements, translation and rotation, are considered to judge whether two correspondences belong to the same cluster or not. As reported in Equations (8) and (9), for two correspondences with rotation and translation matrices $(R_i, T_i), (R_j, T_j)$, if the deviation of rotation and translation matrix $A_r$, $A_T$ are, respectively less than thresholds, they are considered as one cluster; otherwise, they belong to two different clusters. In this study, the rotation threshold is set as 5°, and the translation threshold is 5 mr (mesh resolution, the average distance between points).

$$A_r = \frac{180}{\pi} \arccos \frac{trace(R_i \bullet R_j^T) - 1}{2} \tag{8}$$

$$A_T = \sqrt{(T_i - T_j)^T \bullet (T_i - T_j)} \tag{9}$$

Details of the transformation matrix-based clustering algorithm are shown in the Appendix A, Table A2. Once all pairs are labeled, pairs with the same label belong to one cluster. Finally, the cluster with the maximum size is chosen as the maximum similar set, and other clusters are eliminated (as shown in Figure 7b).

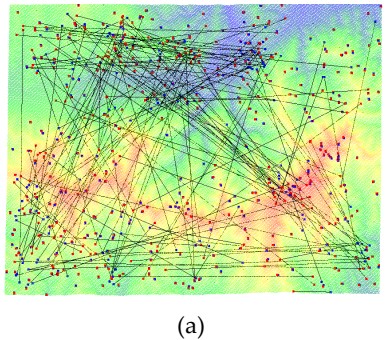 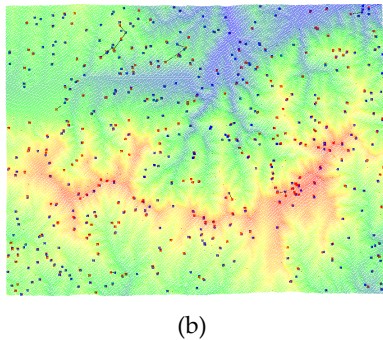

(a)                                                             (b)

**Figure 7.** Clustering based on the transformation matrices. (**a**) is the initial correspondences, (**b**) is the maximum similarity set.

### 3.6. Rodriguez and LS-Based Registration

Once the maximum similar set is obtained, the relationship between source and target can be expressed as Equation (10) [52–54], where $(X_{T_i}, Y_{T_i}, Z_{T_i})^T$ and $(X_{S_i}, Y_{S_i}, Z_{S_i})^T$ are, respectively 3D

coordinates of correspondences in the target and source, $R$ is a rotation matrix and $(T_1, T_2, T_3)^T$ is a translation matrix. Because the number of observations is greater than the number of necessary observations, a least square (LS) algorithm is widely applied to solve parameters. However, because the rotation matrix $R$ is nonlinear in Equation (10), a Gauss–Markov (GM) model is usually established for iterative calculation [34], which is time-consuming. For simplification, a Rodriguez matrix [53,54] is applied instead of the transformation matrix, where the iterative calculation can be effectively avoided.

$$\begin{bmatrix} X_{T_i} \\ Y_{T_i} \\ Z_{T_i} \end{bmatrix} = R \begin{bmatrix} X_{S_i} \\ Y_{S_i} \\ Z_{S_i} \end{bmatrix} + \begin{bmatrix} T_1 \\ T_2 \\ T_3 \end{bmatrix} \tag{10}$$

As reported in Equation (11), a Rodriguez matrix $S$ is a skew-symmetric matrix, which is directly composed of three independent parameters ($a$, $b$, $c$). The Rodriguez matrix has an important character Equation (12), which is greatly helpful when solving parameters. Relationships between the rotation matrix $R$ and the Rodriguez matrix $S$ are reported in Equations (13) and (14). By transforming the rotation matrix $R$ into a Rodriguez matrix $S$, it can effectively transform the observation equation from nonlinear into linear (Equation (15)). The process of the Rodriguez-LS-based method is shown in the Appendix A, Table A3. This method does not involve a solution of trigonometric and anti-trigonometric function in the calculation process, and the calculation accuracy is slightly higher than the traditional seven-parameter estimation methods.

$$S = \begin{bmatrix} 0 & \frac{c}{2} & -\frac{b}{2} \\ -\frac{c}{2} & 0 & \frac{a}{2} \\ \frac{b}{2} & -\frac{a}{2} & 0 \end{bmatrix} \tag{11}$$

$$(I - S)^T = I + S, (I + S)^T = I - S \tag{12}$$

$$\underset{3\times3}{R} = (\underset{3\times3}{I} - \underset{3\times3}{S})(\underset{3\times3}{I} + \underset{3\times3}{S})^{-1} \tag{13}$$

$$R^T = R^{-1} = (I + S)^T (I - S) \tag{14}$$

$$\frac{1}{2}\begin{bmatrix} 0 & -Z_{T_{ji}} - Z_{S_{ji}} & Y_{T_{ji}} + Y_{S_{ji}} \\ Z_{T_{ji}} + Z_{S_{ji}} & 0 & -X_{T_{ji}} - X_{S_{ji}} \\ -Y_{T_{ji}} - Y_{S_{ji}} & X_{T_{ji}} + X_{S_{ji}} & 0 \end{bmatrix}\begin{bmatrix} a \\ b \\ c \end{bmatrix} = \begin{bmatrix} X_{S_{ji}} - X_{T_{ji}} \\ Y_{S_{ji}} - Y_{T_{ji}} \\ Z_{S_{ji}} - Z_{T_{ji}} \end{bmatrix} \tag{15}$$

## 4. Criteria

To quantitatively evaluate the results of the proposed co-registration method, several criteria are applied. The criteria for keypoint extraction, LRF construction, feature-matching and registration are, respectively introduced from Section 4.1 to Section 4.4.

### 4.1. Criteria for Keypoints Extraction

Traditionally, there are two main traits to quantitatively evaluate extracted keypoints: repeatability and distinctiveness. Repeatability is an ability to detect the same keypoints accurately under various nuisances, such as noise corruption, density variation and partial occlusion, while distinctiveness is an ability to be effectively described and matched under wrong correspondences, which is always measured in combination with a specific descriptor [55]. Thus, in this section, only the repeatability is adopted for keypoint evaluation.

The repeatability includes two aspects: absolute repeatability $r_{abs}$ and relative repeatability $r_{rel}$. As reported in Equation (16), a keypoint $p_i$ in the model is said to be repeatable if the distance from its nearest neighbor $q_j$ in the scene is less than a threshold $\varepsilon$ after true transformation ($R_t$, $T_t$). In this

study, this threshold $\varepsilon$ is set to 2 mr. If there are $K$ keypoints extracted in the model, and meanwhile $k$ keypoints are repeatable, then the absolute and relative repeatability can be computed as Equation (17).

$$\left\| R_t p_i^m + T_t - q_j^s \right\| < \varepsilon \tag{16}$$

$$r_{abs} = k, r_{rel} = \frac{k}{K} \tag{17}$$

### 4.2. Criteria for LRF Construction

A stable LRF is of great significance to the distinctiveness and robustness of LRF-dependent descriptors, as its invariance to rigid transformations is achieved by LRF. The direction consistency of LRFs have a greater impact on the description encoding, as the contrary direction of LRFs will result in a completely different coding sequence [44]. Thus, a criterion, direction consistency is defined to judge the error of each coordinate axis.

Let $L_p$ ($L_{p_x}$, $L_{p_y}$, $L_{p_z}$) and $L_q$ ($L_{q_x}$, $L_{q_y}$, $L_{q_z}$) be two LRFs, respectively located on two keypoints $p_i$ and $q_i$ on the source and target. If the distance between two keypoints is less than a threshold $\eta$ (in this study, it is set as 1 mr) after true transformation, then, the dot product of each axis is computed as Equation (18). The X-axis is regarded as consistent if the dot product between two axes $L_{p_x}$ and $L_{q_x}$ is larger than '**0**', which is the same for Y-axis and Z-axis.

$$\cos \theta_x = L_{p_x} * L_{q_x}, \quad s.t. \|p - q\| < \eta \tag{18}$$

$$\cos \theta_y = L_{p_y} * L_{q_y}, \quad s.t. \|p - q\| < \eta \tag{19}$$

$$\cos \theta_z = L_{p_z} * L_{q_z}, \quad s.t. \|p - q\| < \eta \tag{20}$$

### 4.3. Criteria for Feature-Matching

To quantitatively evaluate the performance of the proposed feature-matching method, the number of correct matches $C_p$ is adopted. When the distance $\|q_j - Tp_i\|$ between the keypoint $p_i$ in the source and the keypoint $q_j$ in the target is smaller than a threshold $s$ (7.5 mr in this study), $p_i$ and $q_j$ are considered as a corresponding keypoint pair ($T$ is the true transformation matrix). A match is considered as correct when the corresponding descriptors' keypoints are corresponding keypoint pairs [44].

As one of the widely used measurements in the literature [23,45], the recall and precision are calculated as follows: the precision $P_r$ is calculated as the number of correct descriptor matches with respect to the total number of descriptor matches (Equation (21)), while the recall $R_e$ is the number of correct descriptor matches with respect to the number of corresponding keypoint pairs (Equation (22)). "#" is the total number of samples from a given category.

$$P_r = \frac{\#C_p}{\#\text{descriptor matches}} \tag{21}$$

$$R_e = \frac{\#C_p}{\#\text{corresponding keypoint pairs}} \tag{22}$$

### 4.4. Criteria for Registration

To quantitatively evaluate the performance of the proposed registration method, several criteria are adopted in the experiments: accuracy of rotation matrix $A_r$ (Equation (23)) and ranslation matrix $A_T$ (Equation 24)). Additionally, to assess the performance of gross error elimination, the total positioning error $D_{sum}$ (Equation (25)) and error in X, Y and Z axes $D_x$, $D_y$, $D_z$ of remained matches (Equation (26)) are calculated.

$$A_r = \frac{180}{\pi} \arccos \frac{trace(R_1 \bullet R_2^T) - 1}{2} \tag{23}$$

$$A_T = \sqrt{(T_1 - T_2)^T \bullet (T_1 - T_2)} \tag{24}$$

$$D_{sum} = \frac{1}{n} \sum_{i=1}^{n} \left\| p_i - q_i \right\| \tag{25}$$

$$D_x = \frac{1}{n} \sum_{i=1}^{n} \left\| p_{x_i} - q_{x_i} \right\|, D_y = \frac{1}{n} \sum_{i=1}^{n} \left\| p_{y_i} - q_{y_i} \right\|, D_z = \frac{1}{n} \sum_{i=1}^{n} \left\| p_{z_i} - q_{z_i} \right\| \tag{26}$$

In above equations, $R_2$ and $T_2$ be the calculated rotation and translation matrix, while $R_1$ and $T_1$ are the true rotation and translation matrix; $n$ is the amount of remained correspondences; $\|p_i - q_i\|$ is the distance between the correspondence $p_i$ in the source and $q_i$ in the target after transformation ($R_2$, $T_2$); ($p_x, p_y, p_z$) and ($q_x, q_y, q_z$) are, respectively the x, y and z coordinates of $p_i$ and $q_i$.

## 5. Results

To test the feasibility of the proposed co-registration method, a set of experiments were conducted on three kinds of mimetic satellite LiDAR point clouds and experimental results of the above five steps are, respectively analyzed in follows with the above criteria. Since the performance of the voxel descriptor has been discussed in detail in previous literature [23,32,56], it is not discussed in this study. It is worth noting that when adding noise in following experiments, we add Gaussian noise with a standard deviation of 0.1 mr, 0.2 mr, 0.3 mr and 0.5 mr (mr denotes mesh resolution) to them.

### 5.1. Results of Keypoint Extraction

For extensive, enormous and sparse satellite LiDAR point clouds, a ridgeline-based keypoint extraction is applied, where keypoints are defined as intersections of ridgelines. To reduce the amount of point clouds for subsequent processing, the original point clouds is first, downsampled. Figure 8 shows the results of ridgeline and keypoint extraction. It can be seen that extracted ridgelines can well reflect the topographic characteristics and extracted keypoints, which are colored in blue, are concentrated in the rough area and sparse in the flat area.

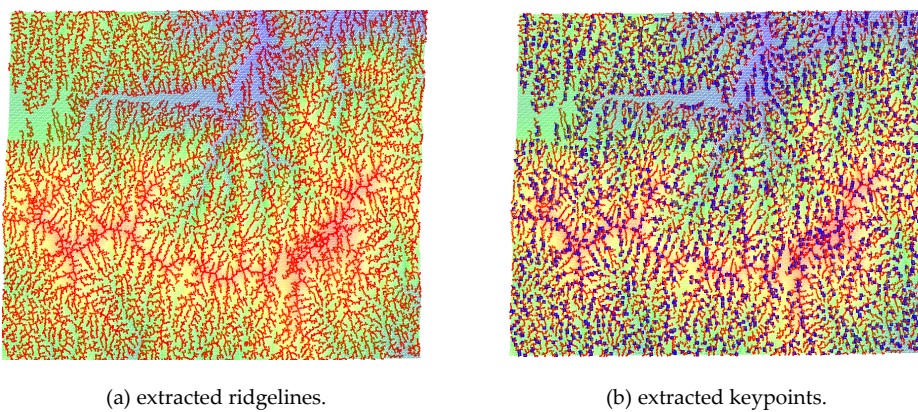

(a) extracted ridgelines.                    (b) extracted keypoints.

**Figure 8.** Results of keypoint extraction.

To get the change in the number of points during this step, the number and efficiency of three point clouds with different resolutions in the same area are recorded as Table 2, where the sampling distance is set as 120 m. It can be seen from table that the number of points of extracted ridgelines is greatly reduced compared with the original point clouds or sampled point clouds. Thus, taking ridgelines as neighborhood information for feature description instead of all point clouds can effectively accelerate the subsequent processing speed. The computational time of ALOS 12 is much higher than others. This is because that a large amount of time is taken to sample. Except for sampling, the time of keypoint extraction of the three data are about 0.3 s.

**Table 2.** Number and efficiency of keypoint extraction.

| Dataset | Original | Sampling | Ridgelines | Keypoints | Efficiency (s) |
|---------|----------|----------|------------|-----------|----------------|
| ALOS 12 | 12,388,722 | 122738 | 24,791 | 2430 | 2.138 |
| ASTER 30 | 2,037,003 | 122650 | 23,140 | 2291 | 0.901 |
| SRTM 90 | 226,426 | 122233 | 24,723 | 2533 | 0.651 |

In addition, the absolute and relative repeatability $r_{abs}$, $r_{rel}$ of keypoints with different Gaussian noise affected are presented as Table 3. It can be seen that, with noise increasing, the relative repeatability of extracted keypoints declines on all three datasets. However, the absolute repeatability of the SRTM 90 decreases first and then increases, while that of ALOS 12 and ASTER 30 decreases all the time. This is because that the point distance of SRTM 90 is much larger than ALOS 12 and ASTER 30. Therefore, the added noise is relatively larger and has a great influence on the local surface, resulting in more points extracted as keypoints.

**Table 3.** Repeatability of keypoints with different levels of noise.

| Data | 0 mr | | 0.1 mr | | 0.2 mr | | 0.3 mr | | 0.5 mr | |
|------|------|------|--------|------|--------|------|--------|------|--------|------|
| | $r_{abs}$ | $r_{rel}$ (%) | $r_{abs}$ | $r_{rel}$ (%) | $r_{abs}$ | $r_{rel}$ (%) | $r_{abs}$ | $r_{rel}$ (%) | $r_{abs}$ | $r_{rel}$ (%) |
| ALOS 12 | 2430 | 100 | 2322 | 95.55 | 2314 | 93.87 | 2286 | 92.88 | 2156 | 87.18 |
| ASTER 30 | 2291 | 100 | 2178 | 86.05 | 2205 | 84.48 | 2041 | 78.31 | 1999 | 73.11 |
| SRTM 90 | 2533 | 100 | 1693 | 51.08 | 2068 | 43.76 | 2352 | 39.53 | 2507 | 37.14 |

*5.2. Results of LRF Construction*

The direction of LRF has a more important impact on feature coding than angle deviation. To evaluate the direction consistency of the constructed LRF, experiments are conducted on the above three datasets with varying noise, where the sources have noise interference, and the targets have no noise interference. For a fair comparison, the direction consistency of LRF decomposed by coordinate covariance and normal vector covariance are both calculated with the same parameters.

Table 4 lists the results of the above three LRF construction methods on three datasets with different levels of noise. It is clear that, from ALOS 0.1 mr to ASTER 0.3 mr, the LRF constructed by coordinate outperforms the LRF constructed by the normal vector. However, with noise continuously increasing, results are on the contrary from SRTM 0.1 mr to SRTM 0.3 mr. It shows that when the noise is large, the LRF constructed by the normal vector is more robust. Additionally, it is clear that the direction of the Z-axis decomposed by coordinate is relatively more stable than the other two axes (Y-axis is determined by X-axis and Z-axis), while the X-axis decomposed by the normal vector is relatively steadier than the other two axes. Thus, by combining two matrices, the proposed method makes full use of the above two methods and the X and Z axes constructed by the proposed method are much more directionally consistent than either.

**Table 4.** Direction consistency of three LRF construction methods.

| Dataset | Coordinate | | | Normal Vector | | | Proposed Method | | |
|---------|------|------|------|------|------|------|------|------|------|
| | X | Y | Z | X | Y | Z | X | Y | Z |
| ALOS 0.1 mr | 2001 | 1997 | 2097 | 2019 | 1867 | 1915 | 2026 | 2034 | 2097 |
| ALOS 0.2 mr | 1925 | 1922 | 2049 | 2014 | 1874 | 1890 | 2013 | 2016 | 2049 |
| ALOS 0.3 mr | 1905 | 1898 | 2018 | 1982 | 1868 | 1897 | 1980 | 1985 | 2018 |
| ASTER 0.1 mr | 1726 | 1721 | 1873 | 1802 | 1568 | 1608 | 1807 | 1812 | 1873 |
| ASTER 0.2 mr | 1414 | 1405 | 1548 | 1516 | 1448 | 1479 | 1522 | 1531 | 1548 |
| ASTER 0.3 mr | 1350 | 1347 | 1479 | 1430 | 1227 | 1263 | 1422 | 1431 | 1479 |
| SRTM 0.1 mr | 528 | 526 | 601 | 577 | 539 | 565 | 584 | 592 | 605 |
| SRTM 0.2 mr | 169 | 165 | 212 | 200 | 183 | 191 | 200 | 204 | 212 |
| SRTM 0.3 mr | 113 | 111 | 156 | 153 | 136 | 141 | 148 | 154 | 156 |

## 5.3. Results of Feature-Matching

To evaluate the proposed MCMF-matching method, a set of experiments are conducted on three simulated satellite LiDAR point clouds with different resolutions in the same area. The results are reported in Table 4. For comparison, experiments are also conducted on the previous KM-matching algorithm, which has shown that it performs better than NN and NNDR-based strategies in previous studies [45]. In Table 5, $K_s$ is the number of keypoints on the source and $K_t$ is the amount of keypoints on the target, while $M_p$ is the amount of matching pairs, $C_p$ is the amount of correct matches, $P_r$ is the precision and $R_e$ is the recall.

**Table 5.** Results of descriptor-matching.

| Dataset | $K_s$ | $K_t$ | KM Algorithm | | | | | MCMF Algorithm | | | | |
|---------|-------|-------|-------|----------|-------|--------|--------|-------|----------|-------|--------|--------|
| | | | $M_p$ | Time (s) | $C_p$ | $P_r$ (%) | $R_e$ (%) | $M_p$ | Time (s) | $C_p$ | $P_r$ (%) | $R_e$ (%) |
| ASTER & ALOS | 2291 | 2430 | 2430 | 5.890 | 647 | 26.6 | 37.7 | 937 | 0.088 | 528 | 56.4 | 30.8 |
| ASTER & SRTM | 2291 | 2533 | 2533 | 7.087 | 419 | 16.5 | 27.1 | 527 | 0.026 | 319 | 60.5 | 20.6 |
| SRTM & ALOS | 2533 | 2430 | 2533 | 3.632 | 557 | 22.0 | 34.9 | 589 | 0.039 | 433 | 73.5 | 27.1 |

It can be seen that the efficiency of the MCMF algorithm is much higher than that of the KM algorithm on above three datasets. This is because the KM algorithm is suitable for square matrices, while MCMF can handle sparse non-square matrices. On one hand, in KM algorithm, when the number of keypoints on the source and target is not equal, keypoints with less amount is expanded by adding some vertices or edges of 0 to make the matching matrix be square, adding much redundant data (for example, (2430–2291) ∗ 2430 in the first data); on the other hand, correspondences of similarity less than a threshold (0.4 in the study), usually do not need to be considered, as it would rarely be a correct match. In this case, a large proportion of matches with small similarity are skipped in MCMF algotirhm, which greatly improve the efficiency. However, some correct correspondences with their similarities less than the threshold are also ignored, resulting in the number of correct matches in the MCMF algorithm is a bit lower than the KM algorithm.

## 5.4. Results of Registration

Random sample-based registration methods (such as RANSAC or 4PCS) are usually used to deal with mismatches. However, these methods are of high randomness, as only three or four correspondences are utilized to determine the final transformation matrix, resulting in many correct matches not participating in it. Therefore, this paper proposes a clustering and Rodriguez-LS-based registration method. To test its applicability, two sets of experiments are carried out: (1) registration with different resolutions in the same area; (2) registration with the same resolution of small overlap.

Figure 9 shows the registered results of the proposed method, where Figure 9a–c is the result of point clouds with different resolutions in the same region and Figure 9d–e is the result of point clouds of small overlap with the same resolution. As can be seen from Figure 9a–c—although details of ridgelines are different due to resolutions (12 m > 30 m > 90 m)—the distribution of extracted ridgelines is basically consistent and all point clouds are well coincided with all mismatches removed. It is obvious from Figure 9d–e that—when the point clouds only partially overlaps—most of the keypoint pairs in the overlapping area are correctly matched and although there is a lack of matches in the right overlapping region in Line 1 & Line 2, both point clouds are well registered no matter on the flat area or rough area.

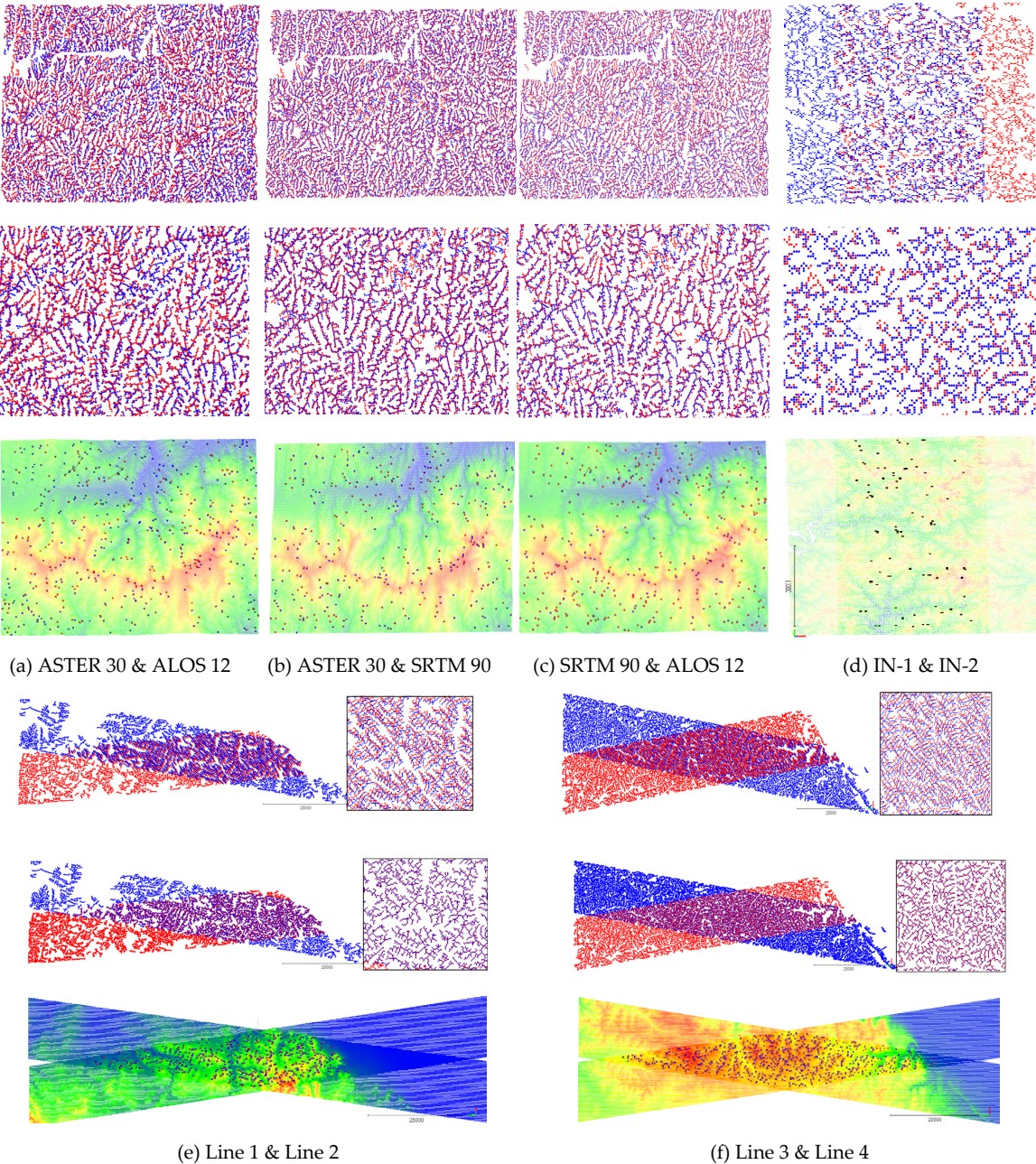

(a) ASTER 30 & ALOS 12    (b) ASTER 30 & SRTM 90    (c) SRTM 90 & ALOS 12    (d) IN-1 & IN-2

(e) Line 1 & Line 2          (f) Line 3 & Line 4

**Figure 9.** Registration of the proposed method. (**a**–**f**) are the registered results of ASTER 30 & ALOS 12, ASTER 30 & SRTM 90, SRTM 90 & ALOS 12, IN-1 & IN-2, Line 1 & Line 2 and Line 3 & Line 4. From the top to bottom are the original ridgelines, registered ridgelines and correspondences distribution.

Table 6 reports the evaluations of above registration results of the proposed method compared with the BSC [44]+KM+RANSAC algorithm, where BSC is a feature descriptor, KM is a feature-matching strategy, and RANSAC is used for registration. It is found that, the rotation and translation deviations of the proposed method are much smaller than the BSC [44]+KM+RANSAC method. There are two reasons: (1) the keypoints we adopted are the intersection of the ridgelines, which is of high uniqueness; the keypoints adopted in the BSC are not specially designed for terrain point clouds and are not distinctive; (2) for the proposed method, after removing mismatches, the remained correspondences of above three datasets are widely distributed in the overlapping area and made full use of for LS-based registration and then the transformation parameters are solved by minimizing the square sum of correspondences' positioning errors; however, the RANSAC algorithm estimates the transformation

matrix by randomly and iteratively selecting three correspondences, which are sparsely distributed in the overlapping area. Positioning errors of correspondences, especially the correspondences of the last iteration, will directly affect the registration accuracy.

**Table 6.** Evaluations of above registration results *.

| Data | The Proposed Method | | | | | | BSC [44]+KM+RANSAC | | | | | |
|---|---|---|---|---|---|---|---|---|---|---|---|---|
| | ① | ② | ③ | ④ | ⑤ | ⑥ | ① | ② | ③ | ④ | ⑤ | ⑥ |
| $A_r$ | 0 | 0 | 0 | 0.2 | 0 | 0 | 9.3 | 36.2 | 13.1 | 238.0 | 974.9 | 0 |
| $A_T$ | 47.1 | 18.3 | 33.1 | 9.9 | 0 | 22.2 | 165.1 | 295.9 | 164.2 | 2857.2 | 19,196.7 | 0 |
| $D_{sum}$ | 66.1 | 80.7 | 66.8 | 56.0 | 1.4 | 15.3 | 87.0 | 234.7 | 126.6 | 511.6 | 556.3 | 1.3 |
| $D_x$ | 51.9 | 67.4 | 39.5 | 6.1 | 0 | 18.1 | 117.3 | 87.9 | 129.5 | 292.9 | 387.8 | 0 |
| $D_y$ | 59.6 | 75.1 | 62.4 | 3.0 | 0 | 14.7 | 79.2 | 192.6 | 120.4 | 478.4 | 517.9 | 0 |
| $D_z$ | 20.3 | 22.9 | 17.0 | 54.9 | 0 | 1.5 | 24.1 | 107.6 | 29.7 | 104.6 | 118.3 | 0 |
| $T_{re}$ | 8.55 | 7.19 | 8.56 | 8.40 | 12.0 | 2.83 | 602.4 | 469.0 | 490.1 | 628.1 | 626.3 | 183.2 |

* The unit of $A_r$ is the degree, while the units of $D_{sum}$, $D_x$, $D_y$, $D_z$, and $A_T$ are meter. $T_{re}$ is the computational time of registration, whose unit is second. For simplicity, we use ①–⑥ to represent ASTER & ALOS, ASTER & SRTM, SRTM & ALOS, Line 1 & Line 2, Line 3 & Line 4 and IN-1 & IN-2, respectively.

It is worth noting that, for Line 1 & Line 2 and Line 3 & Line 4, the registration of BSC [44] + KM + RANSAC method fails because of the small overlapping areas. When the overlapping area is large, such as IN-1 & IN-2, both methods achieve are successful registration results. In terms of efficiency, the proposed method of the above six datasets are much faster than the BSC [44]+KM+RANSAC method. This is because this proposed method only extracts ridgelines as neighborhood feature information, which greatly reduces the number of points processed.

## 6. Discussion

From the above experimental results, it can be seen that registration results are affected by many factors, for example, resolution, noise and topography. The effect of resolution has been shown above through registration results of different resolutions in the same area. In this section, attention will be focused on the influence of different overlap, topography and noise on registration results. As shown in Figure 10, Line 5 & Line 6 and Line 7 & Line 8, Line 9 & Line 10 and Line 11 & Line 12 have the same overlapping areas, but the areas of Line 5 & Line 6, Line 9 & Line 10 are twice that of Line 7 & Line 8 and Line 11 & Line 12.

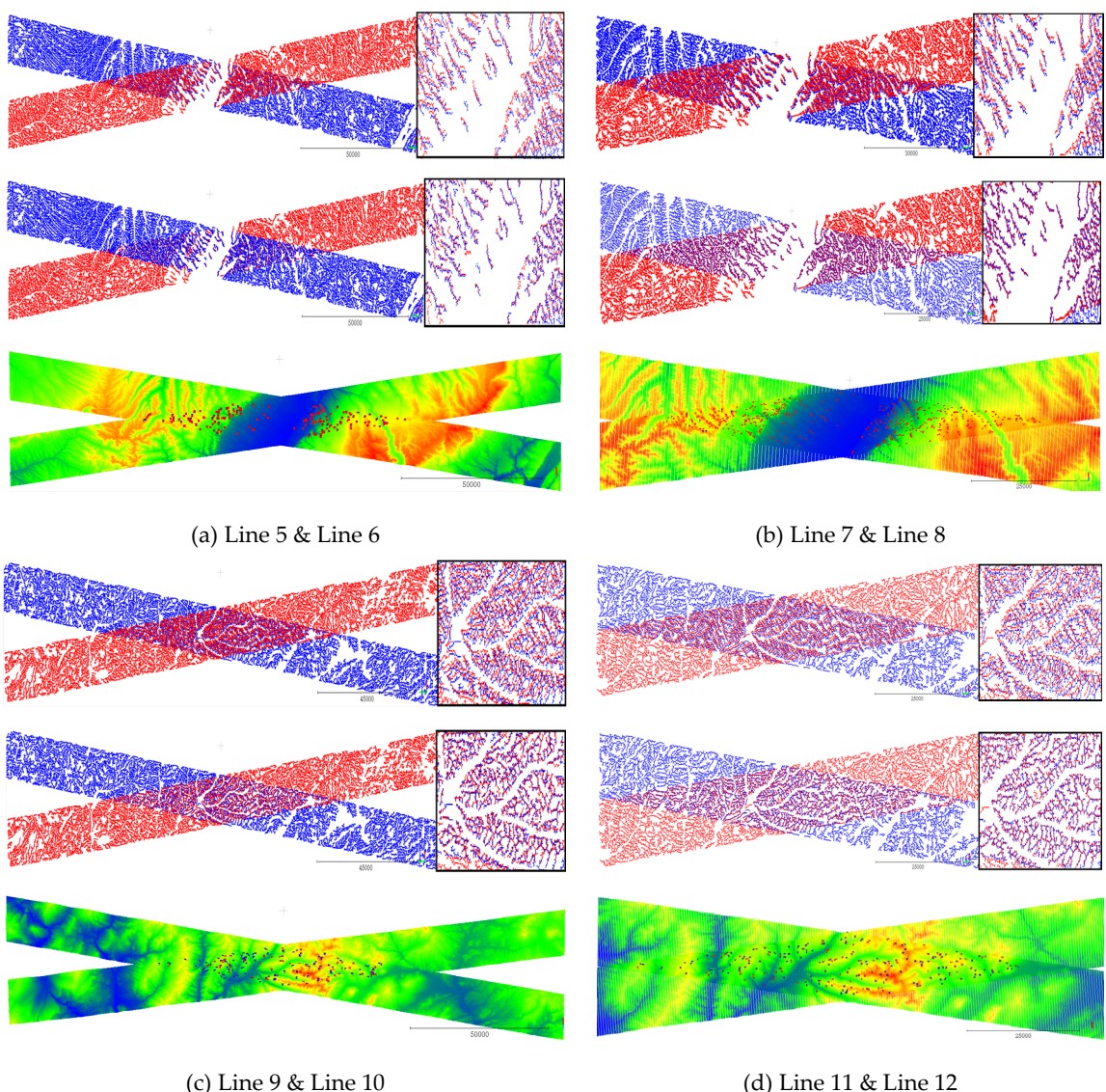

(a) Line 5 & Line 6

(b) Line 7 & Line 8

(c) Line 9 & Line 10

(d) Line 11 & Line 12

**Figure 10.** Registration results with different overlapping rates, topography and noise. From the top to bottom are the original ridgelines, registered ridgelines and correspondences distribution.

## 6.1. Different Overlap

To discuss the influence of different overlap rates on registration results, a set of experiments were conducted. For quantitative assessment, criteria for feature-matching and registration are utilized. Their registration errors are reported in Table 7 compared with the BSC [44]+KM+RANSAC method.

From Table 7, we can see that, for the proposed method, since the above four point clouds could achieve satisfactory registration results through this method, results of larger overlap outperform the smaller ones, especially the Line 7 & Line 8. This is because when the overlap rate is small in Line 5 & Line 6, the proportion of keypoints in the overlapping area to the whole extracted keypoints becomes smaller. Therefore, when feature-matching, keypoints in the overlapping area have more choices and are of higher probability to select mismatches, especially when the features in the overlapping area are not obvious (such as Line 5 & Line 6 and Line 7 & Line 8). When the overlapping rate is large (Line 7 & Line 8), keypoints in the overlapping areas account for more, and their choices are less, resulting in less probability to select mismatches. Thus, the point clouds of large overlap outperforms the small ones. It is worth noting that, if there are more correct pairs in overlap areas participated in the calculation of the rotation matrix, the registration accuracy is

higher, such as Line 7 & Line 8. Due to the almost identical correct matches in Line 9 & Line 10 and Line 11 & Line 12, the registration errors of both two point clouds are actually quite similar. However, for the BSC [44]+KM+RANSAC method, for flat areas (Line 5 & Line 6, Line 7 & Line 8), registration failed regardless of the overlap rates; for rough area and, registrations with high overlap rates (Line 11 & Line 12) are more likely to success compared to samll overlap rates (Line 9 & Line 10).

**Table 7.** Registration errors of four datasets.

| | The Proposed Method | | | | BSC [44]+KM+RANSAC | | | |
|---|---|---|---|---|---|---|---|---|
| | Line 5 & Line 6 | Line 7 & Line 8 | Line 9 & Line 10 | Line 11 & Line 12 | Line 5 & Line 6 | Line 7 & Line 8 | Line 9 & Line 10 | Line 11 & Line 12 |
| $M_p$ | 2377 | 1247 | 546 | 403 | 359 | 238 | 388 | 230 |
| $C_p$ | 159 | 464 | 195 | 204 | 5 | 9 | 13 | 25 |
| $A_r$ (°) | 0.14 | 0 | 0 | 0 | 944.2 | 842.6 | 357.6 | 338.6 |
| $A_T$ (m) | 52.1 | 0.4 | 3.0 | 6.1 | 34,956.4 | 15,269.6 | 10,918.4 | 1271.4 |
| $D_{sum}$ (m) | 158.8 | 2.6 | 29.0 | 34.9 | 728.9 | 845.6 | 795.6 | 362.8 |
| $D_x$ (m) | 142.2 | 2.2 | 62.8 | 61.5 | 429.6 | 508.8 | 462.7 | 361.6 |
| $D_y$ (m) | 91.5 | 1.1 | 17.2 | 22.4 | 552.2 | 794.6 | 753.5 | 141.5 |
| $D_z$ (m) | 114.3 | 0.7 | 15.2 | 15.3 | 398.2 | 169.0 | 243.6 | 307.3 |

### 6.2. Different Topography

The effect of topography on registration results can be mainly divided into two aspects, one is the keypoint extraction, the other is feature-matching, which are both finally reflected in the number of correct correspondences.

As shown in Table 7, for the proposed method, when the overlapping changes, the number of correct matches in Line 5 & Line 6 and Line 7 & Line 8 varies greatly, while that of Line 9 & Line 10 and Line 11 & Line 12 stay stable. This is because of the different topography. The overlapping area of Line 5 & Line 6 and Line 7 & Line 8 is flat (Figure 10a,b), while Line 9 & Line 10 and Line 11 & Line 12 are rough (Figure 10c,d). When the overlap rate is small (Line 5 & Line 6), there are many keypoints extracted. However, becuase their local surface is of low distinctiveness and there are more choices to match, most of keypoints are mismatched (159 correct matches in 2377 matches). When the overlap rate is large (Line 7 & Line 8), the amount of extracted keypoints reduces sharply (from 2377 to 1247). Therefore, keypoints in the overlap area have fewer choices for matching, resulting in more correct matches (from 159 to 464). For Line 9 & Line 10 and Line 11 & Line 12, the change of correct matches is not obvious with the overlap rate varying (from 195 to 204). This is because when the overlap rate varies, although the amount of extracted keypoints changes, the number of matches with similarity greater than the threshold just slightly changes (from 546 to 403) because of their distinct features. However, for the BSC [44]+KM+RANSAC method, registration is often failed when the overlapping area is flat (as Line 5 & Line 6, Line 7 & Line 8), but registration was successful when the terrain is rough.

In short, for the proposed method, when the topography is rough in the overlapping area, the variation of the correct matches is small with the overlap rate changing; when the topography is flat, the number of correct matches fluctuates greatly.

### 6.3. Different Noise

To discuss the impact of noise on registration accuracy, experiments were conducted on Line 11 & Line 12 compared with the BSC [44]+KM+RANSAC method, where Gaussian noise with a standard deviation of 0.1 mr, 0.2 mr and 0.3 mr (mr denotes mesh resolution) is added to the Line 12. To quantitatively assess registration results with varying Gaussian noise, criteria for feature-matching and registration mentioned above are calculated and reported in Table 8.

**Table 8.** Registration errors of point clouds with different noise.

| | The Proposed Method | | | | BSC [44]+KM+RANSAC | | | |
|---|---|---|---|---|---|---|---|---|
| | 0 mr | 0.1 mr | 0.2 mr | 0.3 mr | 0 mr | 0.1 mr | 0.2 mr | 0.3 mr |
| $M_p$ | 403 | 1371 | 1688 | 1763 | 230 | 247 | 247 | 242 |
| $C_p$ | 204 | 192 | 156 | 112 | 25 | 11 | 11 | 7 |
| $A_r$ (°) | 0 | 0 | 0 | 0 | 338.6 | 154.6 | 191.2 | 812.6 |
| $A_T$ (m) | 6.1 | 5.3 | 8.0 | 31.7 | 1271.4 | 3023.9 | 2988.9 | 12,249.5 |
| $D_{sum}$ (m) | 34.9 | 69.3 | 93.1 | 122.8 | 362.8 | 567.2 | 207.3 | 603.7 |
| $D_x$ (m) | 61.5 | 60.0 | 81.1 | 68.0 | 361.6 | 422.1 | 362.4 | 420.5 |
| $D_y$ (m) | 22.4 | 64.4 | 86.9 | 117.6 | 141.5 | 556.4 | 362.5 | 539.3 |
| $D_z$ (m) | 15.3 | 18.7 | 26.1 | 21.9 | 307.3 | 97.1 | 22.6 | 222.6 |

It can be seen that with noise increasing, for the proposed method, the number of matching pairs increases from 403 to 1763, while that of BSC [44]+KM+RANSAC method is almost no change at all. This is because noise makes ridgelines trivial and many points without apparent features are extracted as keypoints in the proposed method, while the BSC [44]+KM+RANSAC method used a ISS (intrinsic shape signature) keypoint extraction, which is not greatly affected by noise. Additionally, for the proposed method, the amount of correct matches gradually reduces from 204 to 112, which is because increasing noise destroyed the original characteristics of ridgelines. Meanwhile, due to the noise disturbed, the position error of correct pairs increases, resulting in the deviations ($D_{sum}$, $D_x$, $D_y$, $D_z$) of correct correspondences raise after registration. However, the error of the transformation matrix, especially the rotation matrix, is essentially consistent, which mainly owes to (1) almost all wrong matches are eliminated by the transformation matrix-based clustering; (2) the optimal transformation matrix is calculated by the Rodriguez-LS algorithm making use of all correct matches. However, for the BSC [44]+KM+RANSAC method, as the noise increases, the registration error becomes larger, and when the noise reached to 0.3 mr, the registration fails. It is worth noting that, due to the large footprint of satellite LiDAR point clouds, the small error of the rotation matrix still has a great impact on the matching deviation, especially on the edge.

## 7. Conclusions

A ridgeline-based terrain co-registration method is proposed to handle extensive, enormous and sparse satellite LiDAR point clouds in rough areas. The presented method showed several merits in experiments: (1) the number of points greatly reduced and keypoints was of high repeatability and distinctiveness; (2) the proposed LRF is of high direction consistency; (3) the MCMF algorithm could efficiently achieve a maximum similarity sum in a sparse-matching graph; (4) the transformtion matrices based clustering makes full use of correct correspondences and can effectively eliminate mismatches. However, there are many aspects that require improvement in future work in order to overcome the limitations of ridgeline-based co-registration. In our cases, the experimental data are simulated from worldwide DEMs and airborne LiDAR point clouds instead of the real satellite LiDAR point clouds, which ignore the impact of platform jitter and atmospheric transmission. Additionally, this method is proposed for rough areas, however, for the flat areas where has no or not enough ridgelines, it will not perform well. Due to the limited research that is reported for registration of satellite LiDAR point clouds, our work may inspire more researchers to work in this field and to achieve better results.

**Author Contributions:** Conceptualization, W.J. and R.Z.; methodology, R.Z.; validation, W.J. and R.Z.; formal analysis, R.Z.; investigation, W.J.; resources, W.J. and R.Z.; data curation, W.J.; writing—original draft preparation, R.Z.; writing—review & editing, R.Z.; supervision, W.J.; project administration, W.J.; funding acquisition, W.J. All authors have read and agreed to the published version of the manuscript.

**Acknowledgments:** The authors would like to express their gratitude to the editors and the reviewers for their constructive and helpful comments for the substantial improvement of this study. This work was supported by the National Key R&D Program of China under Grant 2018YFB0504801.

**Conflicts of Interest:** The authors declare no conflicts of interest.

**Appendix A**

**Table A1.** Algorithm 1 minimum cost–maximum flow-Based Graph-Matching.

| **Input: Similarity Matrix**<br>**Output: Maximum Flow With Minimum Cost** |
| --- |
| (1) initialize the capacity cost graph $G = (V, E, c, w)$; initialize the maximum flow as 0;<br>(2) construct weighted residual digraph;<br>(3) search an augmented path $p$ from **vs** to $v_t$ in the residual digraph. If it is found, return to step 4; otherwise, return to step 6.<br>(4) search for a path with the minimum cost and maximum flow $f_{ij}$ in augmented paths;<br>(5) update the cost and capacity of related arcs in current residual digraph and return to step 3;<br>(6) get a feasible maximum flow $f = \{ f_1, f_2, f_3, \ldots, f_k\}$ with minimum cost, exit. |

**Table A2.** Algorithm 2 Maximum Sample Set Chosen by Clustering.

| **Input: Initial Correspondences with Their Transformation Matrices, $i = 0$**<br>**Output: Cluster with The Maximum Size** |
| --- |
| (1) if the current stack $T$ is empty, select any unmarked matching pair as a seed and put the seed in the stack with a label $i$;<br>(2) pop up the top pair of the stack and look for all unmarked pairs. If the differences between both translation and rotation are less than thresholds, put it into the stack with a label $i$;<br>(3) repeat step (2) until the stack is empty;<br>(1) $i$++; repeat steps (1)–(3) until all pairs are marked. |

**Table A3.** Algorithm 3 Rodriguez-LS-based Registration.

| **Input: Remained Correspondences**<br>**Output: Optimal Transformation Matrix** |
| --- |
| (1) list the observation equation: $\underset{3(n-1)\times 3}{B}\ \underset{3\times 1}{\hat{X}} = \underset{3(n-1)\times 1}{L}$ |
| (2) list the error equation: $\underset{3(n-1)\times 1}{V} = \underset{3(n-1)\times 3}{B}\ \underset{3\times 1}{\hat{X}} - \underset{3(n-1)\times 1}{L}$ |
| (3) calculate $\hat{X}$ according to the principle of least squares: $\underset{3\times 1}{\hat{X}} = \left(\underset{3\times 3(n-1)}{B^T}\ \underset{3(n-1)\times 3}{B}\right)^{-1} \underset{3\times 3(n-1)}{B^T}\ \underset{3(n-1)\times 1}{L}$ |
| (4) calculate the rotation matrix $R$ according to Equation (11); $\underset{3\times 3}{R} = \left(\underset{3\times 3}{I} - \underset{3\times 3}{S}\right)\left(\underset{3\times 3}{I} + \underset{3\times 3}{S}\right)^{-1}$ |
| (5) calculate the translation of all points in the maximum similar set:<br><br>$\begin{bmatrix} T_x & T_y & T_z \end{bmatrix}^T = \dfrac{1}{n}\sum_{i=1}^{n}\left(\begin{bmatrix} X_i^S & Y_i^S & Z_i^S \end{bmatrix}^T - R\begin{bmatrix} X_i^T & Y_i^T & Z_i^T \end{bmatrix}^T\right)$ |

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
