# Peer review of "A Ridgeline-Based Terrain Co-registration for Satellite LiDAR Point Clouds in Rough Areas"

_remotesensing, doi:10.3390/rs12132163_

Round 1

Reviewer 1 Report

This paper presented a pipeline for very-large-area point cloud registration. The developed algorithm seems to be mature. It is tested on synthetic point clouds sampled from satellite DEMS. The experiment and analysis are thorough and sound. The organization of the paper and the writing style is rather clear. There are a few comments which accumulate to "minor revision" in my view. In addition, this manuscript need some proofreading. A few sentences are grammatically incorrect.

My only main concern is on the concept of “satellite point clouds”. In my view, the presented work is about terrain elevation data registration for very large area. Conceptually and methodology-wise, this is more related to satellite DEM registration. The only difference is that the registration is done on discrete point sets, rather on the raster sets. See my later comments for more details.

The tile reads strangely to me. “A …. framework … for ”?

abstract: (4) “differed from RANSAC and 4PCS based methods…” im not sure if this sentence is needed here. there’s no need to mention other specific methods already in the abstract. safely remove this sentence by just saying ”A transformation matrix based clustering …..”

p1 ”To improve the elevation accuracy of satellite images…” check for grammar.

p1. ZY302, icesat2: what about GEDI?

p2: “3D”, “DEM” etc: please define abbreviations at their first appearances. check through the entire manuscript.

p2: I’m not sure about the concept “satellite point clouds” in this paper. Current satellite lidar data are with very sparse density and very large footprint. Whereas in this paper, the registration is done for gridded “dense” point cloud, which is more similar to other terrestrial or aerial lidar products. Especially the coregistration relied on the intersections of ridgelines. I strongly doubt this is well connected to satellite lidar products. Although the authors stated that this is a preparation for future satellite point clouds, do we really foresee “dense gridded satellite point clouds” in the near future? Rather, the methodology presented in paper is more suitable for, and should work well with e.g., satellite DEMs. I’m not sure what’s the real motivation of talking about “satellite point clouds”?

p4 “It is noteworthy that since there is no satellite point cloud up to now…” again I’m confused what is the “satellite point cloud” in this paper?

p4 “where the resolution of the generated point cloud is consistent with the original DEM”: this means one point per cell?

p5 ” strip point cloud”: what is strip point cloud? we need more information about this.

p6 “maximum resolution” what is this value?

p7: if the local surface is projected to XY plane, then only Z axis is used and why the determination of X and Y axes is important in LRF? or does this mean your descriptor is not rotation invariant?

p8: how is the node similarity calculated?

p9: is the clustering done with heuristic test? e.g., all possible corresponding pairs are tested?

table 3 93.87  -> rrel(%)

table 5: how come the amount of matching pairs is larger than the amount of keypoints? there are one-to-many pairs?

p14: wrong figure numbering. please check through the manuscript.

p17: overlaps: “twice range” how are the overlapping rates determined exactly? the width of point cloud strip?

Author Response

Thank you for considering our manuscript for review and giving the opportunity for revision. We appreciate your time to process this manuscript and give review comments immediately.

We have submitted our manuscript entitled, “A Ridgeline-based Co-registration in Preparation for Satellite LiDAR Point Clouds in Rough Areas”, for consideration for publication in journal of Remote Sensing. According to review comments and “Special Instructions to Authors”, we have made some revisions for this manuscript, which mainly focus on experiments and discussion. The details of revisions and responses to reviewers’ comments are listed as following pages. In addition, we polished the manuscript under the help of a native English speaker.

Reviewer 2 Report

This manuscript proposes a novel method for point cloud registration. This method is mainly applicable in rough (mountainous) areas due to the existence of ridgelines, and may not result in satisfactory results in flat areas in case of noisy data or low overlap. Meanwhile, I suggest the authors proofread the manuscript carefully as there are many unclear sentences or grammatical typos in the manuscript. This manuscript does not have line number and makes it difficult to address the line numbers exactly.

Major comments:

Since this method works well especially in rough areas, I suggest you to modify the title accordingly.

Last line in conclusions: “Therefore, a more robust registration method needs to be further studied.” -> so what is the point in using this method? Why do not you use data from  GEDI or other LiDAR sensors to better check the method? For example,  authors could use a dataset from ALS with different conditions (flying height, direction or even platform) along with the current datasets in this paper.

Moderate and Minor comments:

Introduction, line 1: “ Differed from conventional 2D remote” -> “ Different from conventional 2D remote”

Intro, line2 : “ three-dimensional information in high accuracy” ->  use OF or WITH instead of IN

Intro, line 5: “ for example, agriculture, water conservancy”. provide relevant citations here

Intro, “ However, up to now, LiDAR is rarely carried” -> “ However, up to now, LiDAR has rarely been carried”

Intro, “ improve the elevation accuracy of satellite images” -> you mean vertical accuracy?

which satellite images you are talking about? LiDAR images?

Intro,  page 2: “and large scale is a big challenge” -> by scale you mean footprint?

Intro,  page  2: “To study its possibility, this paper”-> “To address this challenge, …”

Section 1.1.1: “RANSAC and its variation” -> “RANSAC and its variations”

section 1.1.1. -> not well written. many sentences are just written without linking them. some sentences are very short and can be combined with others

section 1.1.2 : “4PCS and its variation” -> “4PCS and its variations”

section 1.1.2. :” is proposed aiming at urban” -> “was proposed aiming at urban”

section 1.1.3” LS and its variation”-> “ LS and its variations”

section 1.1.3 :” However, only random errors of target point clouds are considered.for what? error estimation? or what else?

Section 1.1.3 : “(CTLS) is introduced by Chen et” -> “(CTLS) was introduced by Chen et”

Section 1.1.3. : “registration of 3D surface, where an” -> “registration of 3D surfaces, where an”

Section 1.2 “It has two special” -> “they have …”

Section 2 datasets: you have thee sentences about the data you have obtained: “The data can be download from the website” change them to “The data were downloaded from the website”

Section 2. remove descriptions of the SRTM data inside parentheses. You have already mentioned them.

“The SRTM3 DEM (the resolution is approx. 90 meters) is available for over 80% of the globe” > “The SRTM3 DEM is available for over 80% of the globe”

“SRTM1 DEM (approx. 30m resolution) was also produced but is not available for all countries.” -> “SRTM1 DEM was also produced but is not available for all countries.”

Figure 1 caption : “Figure 1. Comparison of different dataset. (a), (b), (c) are respectively the simulated point cloud of SRTM ” -> “Figure 1. Comparison of different datasets. (a), (b), (c) are respectively the simulated point clouds of SRTM ”

Figure 1 , line 2 of caption: “d) and (e) are respectively comparison of details and contours. ” WHAT DO YOU MEAN?

“region. Their range is about 58280.9m * 57504.7m.”-> what  do you mean by range?

“The above three simulated point cloud in Fig” -> “The above three simulated point clouds in Fig”

In table  1, what do you mean by “laser beam”?

Methodology section, “point clouds are composed” -> change ARE to IS

Method section “introduced in follows.” -> “introduced as follows.”

“Meanwhile, great changes will not take place to it” I did not understand what you mean?

What do you mean by AMOUT in “to reduce the amount for subsequent processing”?

What are the exact first two rules in “thinning rules are defined as follows: (1) the former scenic point; (2) the boundary point; (3) the endpoint cannot be deleted; (4) the isolated point cannot be deleted; (5) the connectivity should be unchanged; (6) the edge should be deleted first.”?

Section 3.2 “relatively stabler and unique than other two” -> use more stable instead of stabler

Figure 4. “matrix, (c) is the proposed LRF. ” which one is the proposed LRF in c?

Page 7 line 1: “After a keypoint pi is detected, neighbor points qj in the local surface Si = {qj, || qj - pi || < R} is obtained by a radius R,” -> “Once a keypoint pi is detected, neighbor points qj in the local surface Si = {qj, || qj - pi || < R} are obtained with the radius R,”

what is the differenece between eq 1 and eq 2? you could refer both cov matrixes to a single equation

equations should be numbered in format within  parantheses

Equation 5: Is this true? I assume eq 4 should already be the LRF

Section 3.3. “Thus, a 2D voxelization is applied for feature” -> so you can call it rasterization not voxelization

“the local surface Si = {qj, || qj - pi || < R} is first” -> do not repeat the same set,  simply use S_i

“Thus far, after all voxels in the neighbor are coded” -> use neighbourhood not neighbour

“with a length g * g (” -> “with a length of g * g (”

“Its detailed steps are shown” -> “The detailed steps are shown”

“2. After all pairs are labeled, pairs with the same” -> change AFTER to ONCE

Section 3.6. “After the maximum similar set is obtained, the relationship between source” -> change AFTER to ONCE

“Rodriguez matrix S is an antisymmetric matrix” -> You mean assymetric?

“Rodriguez matrix S are listed in Eq.13 and Eq.14” -> use REPORTED instead of LISTED

Section 4.2. “two aspects: angle deviation and direction consistency” You talk about two parameters but you have not considered ANGLE DEVIATION here?

“two axes Lpx and Lqx is larger than ‘0’” ->  how much larger?

“in the target is smaller than a threshold s, the pi and” -> how much is s? remove THE  before Pi

Section 4.3 ““#” is represented to the total number.” -> “… number of samples from a given category”

Section 4.4 : “accuracy of rotation and translation matrix Ar, AT can” -> “….   matrices, A_r  and A_t respectively,”

“be respectively calculated as Eq.23 and 24. Additionally” -> do not repeat these equations, just refer to them

Sec 5.1 : “5.1 Result of keypoint extraction” ->  “Results of …”

Section 5.1, the first line is very bad written  and should be rewritten again

“table that the point number of extracted ridgelines is” -> What do you mean?

“Besides, as mentioned in section 4.1, the absolute and relative repeatability rabs, rrel of keypoints are calculated as Tab.3” -> “Besides, the absolute and relative repeatability rabs, rrel of keypoints are presented in Tab.3”

“datasets. However, the absolute repeatability of the SRTM 90 increases after decreasing, while ALOS 12 and ASTER 30 both decline.” -> not well written

Correct Table 3: write “rrel (%) “ instead of “93.87”

“with varying noise, where the data with noise is as the source, and the data without noise is as the target.” -> TRUE?

“relatively stabler than the other two axes” ->  use MORE STABLE instead of stabler

“method are much stabler in direction consistency than either.” -> more directionally consistent

Table 4: change “the proposed” to “proposed method”

Section 5.3 “outperforms better than NN and NNDR” -> “performs better than …”

“is the amount of correct matches, Pr is the precision and Cr is the recall.” -?change C_r to R_e

“with similarity less than a threshold” -> “with similarity of less than a threshold”

Section 5.4: “with the same resolution in different bands” -> what do  you mean by bands?

Figure 3 in page 14 should be change to Figure 9!!

Figure 3 “Registration with different resolution in the same ” ->  resolutions

Figure 4 in page 15 should be change to Figure 10!!

Table 6: are these values accuracy or error? I suppose you should change the Table caption

“It can be seen from experiments that, when remained matches are not uniformly distributed and their position errors are small, the transformation matrix estimated by the RANSAC algorithm is slightly better than the proposed method.” -> In real circumstances, I think we face such consition more frequently than when well distributed matches

“The efficiency of each step is listed in Tab.7.” -> reported instead of listed

“time. Especially registration, the time can be almost negligible” -> this  sentence is unclear and should be rewritten

In table 7, you should report the time efficiency of RANSAC too

Table 8: are these accuracy or error?

Table 8: “Table 8. …. of different overlap rate.” -> “…. rates”

Section 6.1. “overlap rate on registration results, a set of experiments are” -> “………… rates ………. was”

“which are 300m in X, Y and Z axes respectively” -> “which was ….”

“Their registration accuracy is listed in Tab.8.” -> “………. Is reported in Table  8”

“From the table, we can see that, since the” -> “From Table 8, we can see that, since the”

“overlap area participated in the calculation of the rotation” -> “overlap areas …..”

“Due to the amount of matching pairs and correct matches in Line 5 and Line 6 are close, the registration accuracy of both” -> “…. And almost identical correct matches in Line 5 and Line 6 , the registration”

“However, due to their local surface is of low distinctiveness” -> “However, because their local surface is of low distinctiveness”

“are mismatched (2218/2377” ->  are these all keypoints mismatched?

“more correct matches (464/1274). For” -> it is not clear what these numbers are referring to. Either mismatched or correct  matches. The two consecutive sentences probably refer to two different things

“Thus, the most similar matches can be found through MCMF matching, so the number of correct matches changes slightly after removing the gross error.” -> so you mean in  rough area, the overlap rate is not crucial and can be as low as possible?

“due to the large range of satellite point cloud, the” -> “due to the large range of satellite point clouds, the”

Table 9: you could compare it with RANSAC under different noise conditions in both flat and rough areas

The first lines in conclusions have been repeated in abstract

“Additionally, the experimental data is simulated from” -> “Additionally, the experimental data are simulated from”

Author Response

(The authors gave the same response as above.)

Reviewer 3 Report

The submitted work presents an automatic co-registration of satellite LiDAR point clouds. Focusing on the ridgelines which play a vital role in terrain representation, the proposed method is especially suitable for areas with great height variations. At the end, experiments have been carried out on synthetic datasets and the results are reported in tables. Generally speaking, the innovation level sounds fine, the structure of the manuscript is acceptable, the goal is interesting and to-the-point, and the text is easy-to-follow but still needs to be checked and proofread.

The main negative point of the submitted work is the lack of comparisons with other state-of-the-art methods. In my opinion, there are some aspects which demand some efforts. Therefore, I suggest addressing the following comments and submitting the revised version of the manuscript to the journal.

  • Although comparisons on different simulated satellite point clouds seem sufficient, the paper lacks of comparisons with some other state-of-the-art methods concerning a similar issue. As a necessary part of the work, I do suggest comparing the achieved results with those of the other competing approaches.
  • There are several other methods which deserve to be addressed/cited in the paper, although not all of them necessarily focused on LiDAR. More specifically, the following works have done valuable research on co-registration of satellite images and could be included in the submitted research:

Han & Oh, 2018, ‘Automated Geo/Co-Registration of Multi-Temporal Very-High-Resolution Imagery’

Barazetti et al, 2014, ‘Automatic Co-registration of Satellite Time Series via Least Squares Adjustment’

Tahoun et al. 2014, ‘Co-registration of Satellite Images Based on Invariant Local Features’

Elliot et al. , 2011, ‘High Resolution Satellite Imagery Coregistration for Accurate Vegetation Change Detection’

Scheffler et al. 2017, ‘AROSICS: An Automated and Robust Open-Source Image Co-Registration Software for Multi-Sensor Satellite Data’

  • Discussing the effect of noise is a valuable part of the work. In practice, however, focusing on only the AWGN is not enough since it is not a realistic assumption in most of the practical applications. I suggest addressing some more realistic types of noises as well.
  • Moreover, the way the levels of additive noises are reported in the text and the tables is not clear enough.
  • Moreover, I suggest introducing the ‘mesh resolution’ (mr) concept in more details to give a better view to the readers.
  • When shortening a word or phrase by means of an abbreviation, please make sure to provide the full term in the text, more specifically before providing the abbreviated form. One example is ‘DEM’ that stands for ‘digital elevation model’.
  • In subsection 1.2. (Contributions), please support the sentence ‘it is universally acknowledged that great changes will not take place to ridgelines generally’ with a reference.
  • I suggest mentioning LiDAR in the abstract as well.
  • Page 2 line 1, the comma after ‘…Altimeter System),’ should be removed
  • Page 2 line 3, I suggest using ‘interest among researchers’ instead of ‘interest in researchers’
  • In the ‘Overview’ subsection, I suggest changing the numbering style from Roman numerals (I, II, III, …) to numbers (1, 2, 3, …)
  • Page 12, I suggest rephrasing “…which is because a…” as well as “This is because that the...”
  • Page 12, The last sentence (above Table 3) is long and not clear. I suggest rewriting the whole sentence in a clear way.

Author Response

(The authors gave the same response as above.)
